# Cost-utility analysis of endoscopic lumbar discectomy following a uniform clinical pathway in the Korean national health insurance system

**Chi Heon Kim**[1,2☯], **Yunhee Choi**[3☯], **Chun Kee Chung**[1,2,4]*, **Seung Heon Yang**[1,2], **Chang-Hyun Lee**[1,2,5], **Sung Bae Park**[1,6], **Keewon Kim**[7,8], **Sun Gun Chung**[7,8]

**1** Department of Neurosurgery, Seoul National University College of Medicine, Seoul, South Korea, **2** Department of Neurosurgery, Seoul National University Hospital, Seoul, South Korea, **3** Division of Medical Statistics, Medical Research Collaborating Center, Seoul National University Hospital, Seoul, South Korea, **4** Department of Brain and Cognitive Sciences, Seoul National University, Seoul, South Korea, **5** Department of Neurosurgery, Seoul National University Bundang Hospital, Seongnam-si, Gyeonggi-do, South Korea, **6** Department of Neurosurgery, Borame Medical Center, Seoul National University Boramae Hospital, Seoul, South Korea, **7** Department of Rehabilitation Medicine, Seoul National University College of Medicine, Seoul, South Korea, **8** Department of Rehabilitation Medicine, Seoul National University Hospital, Seoul, South Korea

☯ These authors contributed equally to this work.
* chungc@snu.ac.kr

## Abstract

## Introduction

Full-endoscopic lumbar discectomy (FELD) is a type of minimally invasive spinal surgery for lumbar disc herniation (LDH). Sufficient evidence exists to recommend FELD as an alternative to standard open microdiscectomy, and some patients prefer FELD due to its minimally invasive nature. However, in the Republic of Korea, the National Health Insurance System (NHIS) controls the reimbursement and use of supplies for FELD, but FELD is not currently reimbursed by the NHIS. Nonetheless, FELD has been performed upon patients' request, but providing FELD for patients' sake is inherently an unstable arrangement in the absence of a practical reimbursement system. The purpose of this study was to conduct a cost-utility analysis of FELD to suggest appropriate reimbursements.

## Method

This study was a subgroup analysis of prospectively collected data including 28 patients who underwent FELD. All patients were NHIS beneficiaries and followed a uniform clinical pathway. Quality-adjusted life years (QALYs) were assessed with a utility score using the EuroQol 5-Dimension (EQ-5D) instrument. The costs included direct medical costs incurred at the hospital for 2 years and the price of the electrode ($700), although it was not reimbursed. The costs and QALYs gained were used to calculate the cost per QALY gained.

**Data Availability Statement:** All relevant data are within the paper and its Supporting Information files.

**Funding:** This work was supported by Doosan Yonkang Foundation (800-20210527). This study was supported by Seoul National University Hospital research fund (0420210540). A grant from the Korea Health Technology Research & Development Project supported this study through the Korea Health Industry Development Institute (KHIDI) funded by the Ministry of Health & Welfare, Republic of Korea (HC15C1320). The funders had no role in study design, data collection and analysis, decision to publish, or preparation of the manuscript

**Competing interests:** The first author (CHK) is a consultant of RIWOspine GmBH. All the authors declare that they have no conflicts of interest concerning the materials/methods used in this study or the findings described in this paper. This does not alter our adherence to PLOS ONE policies on sharing data and materials.

## Result

Patients' mean age was 43 years and one-third (32%) were women. L4-5 was the most common surgical level (20/28, 71%) and extrusion was the most common type of LDH (14, 50%). Half of the patients (15, 54%) had jobs with an intermediate level of activity. The preoperative EQ-5D utility score was 0.48±0.19. Pain, disability, and the utility score significantly improved starting 1 month postoperatively. The average EQ-5D utility score during 2 years after FELD was estimated as 0.81 (95% CI: 0.78–0.85). For 2 years, the mean direct costs were $3,459 and the cost per QALY gained was $5,241.

## Conclusion

The cost-utility analysis showed a quite reasonable cost per QALY gained for FELD. A comprehensive range of surgical options should be provided to patients, for which a practical reimbursement system is a prerequisite.

## Introduction

Lumbar disc herniation (LDH) is one of the most common causes of low back and leg pain; surgery for LDH is recommended in fewer than 10% of patients, when the pain is medically intractable for at least 4–6 weeks or a neurological deficit such as weakness accompanies the condition [1]. In recent years, minimally invasive surgery, including full-endoscopic lumbar discectomy (FELD), has emerged as a surgical option of interest for both patients and surgeons [2–7]. The top 100 most-cited publications on spinal endoscopic surgery are from the USA (49), Germany (19), Republic of Korea (ROK) (15), Japan (6), China (5), Italy (2), Australia (1), France (1), Canada (1) and the Netherlands (1) [3]. As these numbers show, the ROK is among the leading countries regarding research and clinical practice in this field. When a surgical technique is evaluated, a cost-utility analysis should be performed to evaluate costs and benefits to the patient and to assess the economic burden to society [8–10]. This issue is particularly important for countries with national health insurance systems [8, 11, 12]. All people in the ROK are beneficiaries of the National Health Insurance System (NHIS) and all hospitals follow the regulations of the NHIS. LDH accounts for a large proportion (25%, $980,878/year) of medical expenditures for spinal degenerative disease in the ROK, and any increase in those expenditures would impose a major burden on the NHIS [5, 7, 13–15]. Previous studies have shown that endoscopic lumbar discectomy was cost-effective compared to microdiscectomy [8, 12, 16]. Recent systematic reviews have shown superior outcomes of endoscopic surgery compared to conventional open discectomy, especially with regard to disability, duration of surgery, overall complications, and length of hospital stay [17, 18]. A recent randomized controlled multi-center trial by Gadjradj et al. showed better clinical outcomes after FELD than after open discectomy, and the cost of surgery was lower with FELD than with open discectomy [4, 16]. Economic acceptability is usually judged by willingness to pay (WTP), and those studies showed the WTP for FELD [4, 16]. The results enabled FELD to be included in the health insurance package of the Netherlands [4, 16]. Despite these results supporting FELD, statistics from national registry data have demonstrated that the proportion of FELD decreased from 17% in 2003 to 5% in 2008, while the number of patients with LDH increased by 2.3-fold [12]. In 2020, the proportion of FELD was 10.7% (8,818/82,338), while that of open discectomy was 89.3% (73,520/82,338) (http://opendata.hira.or.kr).

The outcomes of open discectomy and endoscopic discectomy are similar, but physicians should inform patients about all possible options and respect their choices. Many patients choose to receive minimally invasive surgery if possible, expecting less postoperative pain, a shorter recovery time, a lower risk of complications, less need for painkillers, and better cosmetic outcomes, and those expectations are fulfilled by FELD [4, 7, 14, 15, 18–20]. Endoscopic spinal surgery is costly, considering essential supplies such as an endoscope, camera, endoscopic drill, endoscopic forceps, and electrode, but the cost of those supplies are not reimbursed by the Korean NHIS and the burden of the costs is on the hospital and surgeons. The electrode is an especially essential tool in FELD. It is specifically designed for the surgeon to navigate the tip of the electrode into the spinal canal through a spinal endoscope, and it is used for coagulation, shrinkage of tissue, and dissection. The choice among surgical options would be fairer if the actual costs of those supplies were reimbursed, but WTP for FELD has not yet been determined in the ROK [8–10]. To address this gap, we provided a cost-utility of FELD and suggested an appropriate cost for reimbursement of the supplies used in FELD [11].

## Materials and methods

This study was a subgroup analysis of a previous comprehensive cohort study [21]. The previous study aimed to compare non-surgical and surgical outcomes of lumbar disc herniation (LDH) in patients who voluntarily visited a clinic for a second opinion after surgery was recommended by another physician who actively treats spinal disease (i.e., a spinal specialist physician). As a cohort study, participants were included in either the non-surgery or surgery cohort [21]. The comprehensive study included 128 cases (surgery cohort, n = 57; non-surgery cohort, n = 71) (Fig 1).

All surgeons had more than 5 years of experience with either standard open microscopic discectomy or FELD [22–25]. The detailed standard surgical procedures were shared by all surgeons and researchers, and each surgeon was asked to operate using the routine surgical technique with which the surgeon was most confident. The surgery cohort included 34 cases of FELD and 23 cases of open microscopic discectomy. FELD and open discectomy were performed in different hospitals. Patients with FELD followed clinical pathway during admission, but patients with open discectomy did not follow clinical pathway. After surgery, the patients were scheduled to visit the outpatient clinic at 1, 3, 6, and 12 months and yearly thereafter. For the present study, patients with complete records of hospital costs and clinical outcomes at 2 years were selected. At 24 months, 28 patients with FELD and 15 patients with open discectomy had a complete record of hospital costs and those were included in the present analysis (Fig 1). The study was approved by the institutional ethical review board (H 1605-013-759) and registered at both clinicaltrials.gov (NCT02883569, first posted on Aug/30/2016) and the Clinical Research Information Service (https://cris.nih.go.kr/cris/en/) (KCT0000203). All research was performed in accordance with the relevant laws/guidelines/regulations of the Republic of Korea, and the present study was conducted in accordance with the principles of the Declaration of Helsinki. Written informed consent was obtained from all participants and/or their legal guardians.

### Clinical pathway of endoscopic lumbar discectomy

All patients with FELD were managed with the same protocol and orders (clinical pathway), which were developed by the clinicians [26]. When the decision was made to perform surgery, basic preoperative check-ups such as blood tests, electrocardiography, chest X-rays, and spinal X-rays were performed at the outpatient clinic. On day 1 (preoperative day 1), the patient was admitted in the afternoon for a preoperative check-up by an anesthesiologist and other

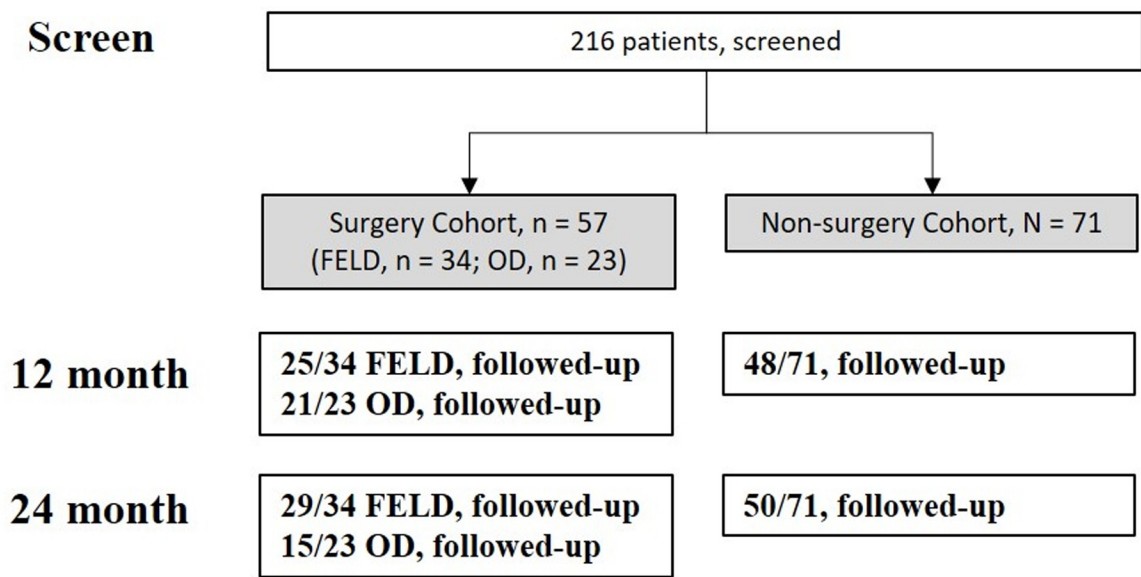

**Fig 1. Patient flow chart.** Initially, 216 patients were screened and 128 cases (surgery cohort, 57; non-surgery cohort, 71) were included in this comprehensive cohort study. The number of patients at each follow-up period is described in the boxes. Abbreviations: FELD, full-endoscopic lumbar discectomy; OD, open microdiscectomy.

physicians, and provided informed consent for surgery. On day 2 (the day of surgery), the patient was transferred to the operation room after a sensitivity test for antibiotics (first-generation cephalosporin) and underwent surgery under general anesthesia. A Foley catheter was not inserted. After the operation, the patient was moved to the post-anesthesia care unit and was returned to the general ward after fully awakening, as confirmed by an anesthesiologist. The patient was encouraged to stand up and walk around the hall of the general ward with a walking aid within 1 hour of returning to the general ward. Pain was controlled with oral acetaminophen and a regular diet was allowed in the evening after confirmation of bowel sound or passing flatus sounds. Postoperative magnetic resonance imaging (MRI) was taken during the evening or night to check the results of discectomy. On day 3 (postoperative day 1), the patient was discharged home in the afternoon after confirmation of independent ambulation, improvements of preoperative symptoms/signs and the absence of any postoperative complication such as a neurological problem. After discharge, a lumbar-supporting brace was not applied, and office work was encouraged within 1 month after surgery. However, strenuous activities such as sports, vigorous leisure activities, and weightlifting were not allowed until 3 months after surgery. The patient was scheduled to visit the outpatient clinic at postoperative 1, 3, 6, and 12 months and yearly thereafter. If pain recurred, it was managed with drugs and/ or epidural injections on an outpatient basis. If the pain was unbearable or those measures were not effective for 2 months, MRI was recommended to check for the possibility of recurrence.

## Surgical procedure

All operations were performed in the prone position under general anesthesia. The FELD procedures were performed as previously described [2, 21, 23, 27]. Two approaches were used according to the level and type of LDH. Generally, transforaminal endoscopic lumbar discectomy (TELD) was used for LDH located at L4-5 or above, and interlaminar endoscopic lumbar discectomy (IELD) was used for LDH at L5-S1 [28, 29]. IELD was specifically selected for

highly migrated LDH or huge LDH occupying more than 50% of the spinal canal [27–30]. The open discectomy was performed under general anesthesia with approximately 2.5cm skin incision. The surgical corridor was developed with an expandable tubular retractor and the standard microsurgical techniques were used for discectomy.

## Clinical parameters

All participants were asked to complete patient-reported outcome questionnaires before the operation and at postoperative 1, 3, 6, 12, and 24 months. The questionnaires included numeric rating scores (NRS) for the back (NRS-B, x/10) and leg (NRS-L, x/10), the Korean version of the Oswestry Disability Index (K-ODI, x/100), [31] quality of life measurements from the EuroQol 5-Dimension instrument (EQ-5D, https://euroqol.org/eq-5d-instruments/eq-5d-5 l-about/), and the EQ visual analogue scale (EQ-VAS). Utility scores were calculated from EQ-5D and were normalized to a range from "health worse than death," represented by a score of 0, to "perfect health," represented by a score of 1 [32]. The EQ-VAS, which describes perceptions of health, was scaled from 0 (worst health) to 100 (best health). Occupational activity (OA) was classified into three categories: high OA, intermediate OA, and low OA [33].

## Costs

All included patients followed the clinical pathway described above, and all were beneficiaries of the NHIS. The total cost included both the price of the electrode ($700) and direct medical costs, such as the cost of the operation ($625), general anesthesia, the use of drugs, procedures at the bedside such as dressings, the room ($292 for a single room, $167 for a double room and $17 for a shared room per night), meals, imaging studies such as X-rays and magnetic resonance imaging ($667), supplies for dressings, and costs for outpatient clinic visits [34]. The costs of the supplies used for FELD, such as the electrode and drill, and the initial cost of the installation of the endoscopic system were not reimbursed by the NHIS. The price of the electrode was included in the present study, because the electrode is an essential supply for FELD and the purpose of this study was to suggest a practical system of reimbursement for supplies.

We selected patients from one hospital because costs may not be uniform across hospitals [8]. The hospital analyzed in the present is a tertiary referral hospital that strictly follows the regulations of the NHIS. The costs incurred during admission for surgery and at outpatient clinics were retrieved from the hospital records and presented based on the timing of outpatient clinic visits. The costs during the first 1 month included costs incurred upon admission for surgery.

## Statistical analysis

The characteristics of patients are described as mean ± standard deviation for continuous variables and frequency (proportion) for categorical variables. A generalized linear mixed-effect model was utilized to evaluate changes in clinical outcomes and total costs over time: before surgery and 1, 3, 6, 12, and 24 months after surgery. In the models, the time when the clinical outcome was obtained was defined as the fixed effect and the patients as the random effect. If the overall time effect was significant in the mixed effect model ($p < 0.05$), then the difference from the baseline values was estimated with 99% confidence intervals and tested using the 1% level of significance by the Bonferroni method.

The cohort was followed up for 2 years and the average utility score of the EQ-5D during 2 years after FELD was estimated from the linear mixed-effect model. The quality-adjusted life years (QALYs) gained over 2 years after FELD were calculated using the following formula:

QALYs gained = Years of life * difference of utility score before and 2 years after FELD

The cost per QALY gained was calculated using the following formula.

$$\text{Cost per QALY gained} = \frac{\text{Cost}}{\text{QALY gained}}$$

All statistical analyses were performed using SAS version 9.4 (SAS Institute, Cary, NC, USA).

## Results

The characteristics of the patients are summarized in Table 1. Patients' mean age was 43 years and one-third (32%) were women. L4-5 was the most common surgical level and protrusion and extrusion comprised 90% of the cases. More than half of the patients (54%) had an intermediate level of OA, and 30% of the patients were smokers (Table 1).

Table 2 shows the estimated mean and the difference of clinical values between before and after FELD. Postoperatively NRS-B, NRS-L, K-ODI, and EQ-5D VAS significantly improved (Table 2). Those parameters showed significant improvement at postoperative 1 month ($p < 0.05$), and this improvement was maintained during follow-up (Table 2). The EQ-5D utility score improved 1 month postoperatively and then remained stable during the follow-up

**Table 1. Characteristics of patients.**

| Variable | N = 28 |
|---|---|
| Age, mean ± SD | 43.04±11.23 |
| Female, n (%) | 9 (32%) |
| Weight (kg) | 73.23±13.38 |
| Height (cm) | 170.74±8.8 |
| BMI (kg/m$^2$) | 25.0±3.49 |
| Levels of herniation | |
| L3-4 | 1 (4%) |
| L4-5 | 20 (71%) |
| L5-S1 | 7 (25%) |
| Type of herniation | |
| Protrusion | 11 (39%) |
| Extrusion | 14 (50%) |
| Sequestration | 3 (11%) |
| Occupational activity | |
| High | 7 (25%) |
| Intermediate | 15 (54%) |
| Low | 6 (21%) |
| Diabetes, n (%) | 1 (4%) |
| Hypertension, n (%) | 2 (7%) |
| Smoking, n (%) | 8 (29%) |
| NRS-B, mean ± SD | 5.89 ± 2.27 |
| NRS-L, mean ± SD | 6.54 ± 1.99 |
| K-ODI | 50.16±17.47 |
| EQ-5D utility score | 0.48±0.19 |
| EQ-VAS | 46.96±18.43 |

Abbreviations: BMI, body mass index; NRS-B, numerical rating scale for back pain; NRS-L, numerical rating scale for leg pain; K-ODI, Korean version of the Oswestry Disability Index; EQ-5D, quality of life measurements from the EuroQol 5-Dimension instrument; EQ-VAS, visual analog score from the EuroQol 5-Dimension instrument.

**Table 2. Clinical outcomes and cumulative total costs.**

| Mean (99% confidence interval) | Pre-operation | 1 month | 3 months | 6 months | 12 months | 24 months | p-value† |
|---|---|---|---|---|---|---|---|
| K-ODI | 50.16 (43.11, 57.22) | 24.76 (17.71, 31.82) | 16.49 (9.11, 23.87) | 14.31 (6.31, 22.31) | 13.98 (5.98, 22.0) | 11.04 (3.69, 18.40) | < .0001 |
| Difference from baseline‡ | | 25.40 (18.89, 31.89) | 33.67 (26.84, 40.51) | 35.84 (28.33, 43.36) | 36.18 (28.67, 43.69) | 39.11 (32.29, 45.93) | |
| p-value* | | < .0001 | < .0001 | < .0001 | < .0001 | < .0001 | |
| NRS-back | 5.89 (5.07, 6.72) | 2.57 (1.75, 3.39) | 2.04 (1.15, 2.92) | 2.03 (1.03, 3.04) | 1.78 (0.78, 2.79) | 1.7 (0.82, 2.58) | < .0001 |
| Difference from baseline | | 3.32 (2.29, 4.35) | 3.86 (2.78, 4.94) | 3.86 (2.68, 5.04) | 4.11 (2.93, 5.29) | 4.2 (3.12, 5.27) | |
| p-value* | | < .0001 | < .0001 | < .0001 | < .0001 | < .0001 | |
| NRS-leg | 6.54 (5.69, 7.39) | 2.43 (1.58, 3.28) | 1.76 (0.85, 2.66) | 1.66 (0.64, 2.68) | 1.7 (0.68, 2.72) | 1.44 (0.53, 2.34) | < .0001 |
| Difference from baseline | | 4.11 (3.12, 5.1) | 4.78 (3.74, 5.82) | 4.88 (3.74, 6.02) | 4.83 (3.69, 5.97) | 5.1 (4.06, 6.14) | |
| p-value* | | < .0001 | < .0001 | < .0001 | < .0001 | < .0001 | |
| EQ-5D utility score | 0.48 (0.41, 0.54) | 0.75 (0.69, 0.82) | 0.8 (0.73, 0.87) | 0.83 (0.75, 0.9) | 0.85 (0.77, 0.92) | 0.87 (0.8, 0.94) | < .0001 |
| Difference from baseline | | -0.27 (-0.35, -0.2) | -0.32 (-0.4, -0.24) | -0.35 (-0.43, -0.26) | -0.37 (-0.45, -0.28) | -0.39 (-0.47, -0.31) | |
| p-value* | | < .0001 | < .0001 | < .0001 | < .0001 | < .0001 | |
| EQ-VAS | 46.96 (39.84, 54.09) | 73.93 (66.8, 81.06) | 79.9 (72.34, 87.45) | 78.84 (70.4, 87.27) | 82.1 (73.67, 90.53) | 82.16 (74.61, 89.71) | < .0001 |
| Difference from baseline | | -26.96 (-34.83, -19.1) | -32.93 (-41.19, -24.68) | -31.87 (-40.94, -22.81) | -35.13 (-44.19, -26.07) | -35.19 (-43.44, -26.95) | |
| p-value* | | < .0001 | < .0001 | < .0001 | < .0001 | < .0001 | |
| Cumulative total cost ($) | | 3,255 (2,974, 3,536) | 3,409 (3127, 3,691) | 3,425 (3,144, 3,706) | 3,448 (3,167, 3,729) | 3,459 (3,174, 3,745) | 0.01 |

Abbreviations: EQ-5D, quality of life measurements from the EuroQol 5-Dimension instrument; NRS-back, numeric rating scale of back pain; NRS-leg, numeric rating scale of leg pain

‡ The difference from baseline is defined as 'before FELD–after FELD.'

* P-value for difference from baseline was adjusted by the Bonferroni method for testing at multiple time points.

† P-value for overall time effect

**Table 3. Itemized costs at postoperative 1 month.**

| Itemized cost | Cost ($) |
|---|---|
| Surgery | 625 |
| Electrode | 700 |
| Anesthesia | 155 |
| Room (mean ± SD) | 95±97 |
| Single* | 3 |
| Twin* | 10 |
| Shared* | 15 |
| MRI (mean ± SD) | 867±311 |
| Pre-op* | 7 |
| Post-op* | 28 |
| Meal | 16 |
| Total cost, mean (95% CI)† | 3,255 (2,974, 3,536) |

*Number of patients.

SD, standard deviation; MRI, magnetic resonance imaging; Pre-op, preoperative; Post-op, postoperative.

†Total included the costs of itemized costs and other non-itemized costs such as general care and nursing.

period. The average EQ-5D utility score during 2 years after FELD was estimated as 0.81 (95% CI: 0.78–0.85); thus, 1.62 QALYs were obtained over 2 years after FELD. The number of QALYs gained over this 2-year period was 0.66 (Table 2). The total cost for 2 years were $3,459 (95% CI, $3,174–$3,745). Most costs were incurred during the first month postoperatively (94%, $3,255/$3,459), which included the costs for FELD and the electrode, while the remaining costs included the costs of outpatient clinic visits and X-rays during follow-up (Table 3). Consequently, the cost per 1 QALY gained was $5,241 (dollar/QALY gained).

## Discussion

The purpose of this study was to present a cost-utility analysis of FELD in order to suggest a reasonable amount of reimbursements for FELD supplies [11]. According to data from 2016, 7,895,929 patients were diagnosed with a spinal disease out of the 52 million people in the ROK [13]. LDH was the second most common diagnosis (3,779/100,000) after nonspecific low back pain [13]. In recent years, the number of patients with spinal disease has increased, along with the expenses for spinal disease [13]. Considering that about 10% of LDH cases require surgical treatment, the burden of surgery for LDH is heavy for the NHIS [13].

In evaluating a new surgical technique, the incurred cost to improve a unit in health outcome measurement is a common method in evaluating the technique under the health insurance system [35, 36]. Utility measures varied between the EQ-5D and variations of the Short-Form Health Survey [35]. Effect measures varied widely between Visual Analogue Scale for pain, Neck Disability Index, Oswestry Disability Index, reoperation rates and adverse events [35]. The current study used EQ-5D for cost-utility analysis. The analysis showed the cost to improve a perfect 1-year quality of life per person (quality adjusted life year, QALY), which was the most common measurement for cost-utility analysis.

In the USA, surgery showed better functional outcomes than non-surgical treatment in the Spine Patient Outcomes Research Trial (SPORT) [1, 37]. According to the SPORT study, surgery costed more than non-surgical interventions ($27,273 vs. $13,135), but surgery showed a lower cost per QALY gained than non-surgical interventions ($34,355 vs. $69,403) [38]. Consequently, early surgical treatment was recommended, considering that the WTP per 1 QALY gained was between $50,000 and $100,000 in the USA [38–40]. In the ROK, not all costs for non-surgical treatment are covered by the NHIS and a fair cost-effectiveness comparison with surgical treatment, which is covered by the NHIS, is not possible. When surgery is necessary, patients usually hope to minimize surgical treatment because of concerns such as tissue injury, recurrence, hospital stay, and sick leave [21, 41]. Although FELD is not perfect, it is able to address these issues [18]. Over the course of the history of endoscopic spine surgery, which has lasted for more than 30 years, many countries have adopted endoscopic spine surgery for these reasons, and its effects have been verified in many studies and systematic reviews [2, 3, 8, 12, 17, 18, 24]. Recently, Gadjradj et al. published the results of a randomized controlled multi-center trial comparing FELD and open microdiscectomy [4]. At 12 months, patients who were randomized to FELD had a statistically significantly lower visual analogue scale score for leg pain (median 7.0, interquartile range 1.0–30.0) compared with patients randomized to open microdiscectomy (16.0, 2.0–53.5) (between-group difference, 7.1; 95% CI, 2.8–11.3) [4]. Other parameters, such as ODI, back pain, health-related quality of life, and self-perceived recovery, favored FELD. In addition, FELD was more cost-effective than open microdiscectomy [16].

In this regard, surgery would be beneficial both from the patient's perspective and from an economic standpoint when it is necessary, and FELD would be an attractive option. Because the ROK is one of the leading countries in FELD, the time has come to scrutinize the economic aspects of FELD to see whether there is room for actual reimbursements by the NHIS [3, 28,

29, 42, 43]. Choi et al. performed a cost-utility analysis comparing FELD and open microdiscectomy and showed that the costs per 1 QALY gained were $24,874 and $28,226 for TELD and IELD, respectively, while the cost per 1 QALY gained was $34,840 for open microdiscectomy [8]. They included indirect medical costs, direct medical costs, and price of the electrode and drills, even though the costs of those supplies were not charged to the patients, hoping that the data could be referenced by the NHIS when deciding upon an affordable reimbursement [8]. FELD saved $8,063 per 1 QALY gained compared with microdiscectomy [8]. Kim et al. also showed that endoscopic spinal surgery was more cost-effective than open microdiscectomy using the NHIS database [12]. However, offering FELD imposed a major economic burden on hospitals and surgeons, because the costs of endoscopic supplies, such as electrodes and drills, as well as the initial installation of the FELD system, were not reimbursed by the NHIS. Considering the cost of supplies ($700–$1,200 for the electrode and $300 for the drill) per surgery and the initial cost of installing the FELD system (around $150,000), performing FELD with a surgical fee of around $1,000 without reimbursement for the supplies has caused an economic deficit for hospitals and doctors.

Japan has a similar national health insurance system to that of the ROK and there are many active endoscopic surgeons in Japan [44]. Japan also shares the issue of practical reimbursement for endoscopic spine surgery, which requires supplies such as endoscopes, endoscopic drills, and electrode [11]. The cost of supplies was (FFE5) 158,569 ($1,459), including the cost of the electrode ((FFE5)87,912, $808), drill burr ((FFE5)29,912, $275), and the maintenance cost of the endoscope and forceps ((FFE5)40,475, $372) [11]. Moreover, an endoscopic system must be installed to perform endoscopic surgery, which costs (FFE5)11,035,000 ($101,522). However, the surgical procedure and reimbursement fee per case is (FFE5)303,900 ($2,796) [11]. Hospitals cannot make ends meet with the current reimbursement system [11]. Nonetheless, endoscopic surgery has been performed for the sake of patients [3, 11]. Although supplies are partially reimbursed, endoscopic spinal surgery is not an economic option in Japan. Recently, Gadjradj et al. showed better leg pain and QALYs after endoscopic lumbar discectomy than after open discectomy [4, 16]. through a randomized controlled trial. This study showed that the WTP for endoscopic lumbar discectomy will be similar or higher than the WTP of 6112 Euros ($6612) in the Netherlands [4, 16]. Those results prompted the Dutch Ministry of Health to include endoscopic lumbar discectomy in the basic health insurance package [4, 16]. In the ROK, the cost of supplies is not reimbursed by the NHIS and the surgical fee is one-third of that in Japan, but the economic burden has fallen on hospitals and doctors. The ROK ranked 10th in the 2020 global GDP ranking, and its GDP per capita is more than $30,000. Usually, the WTP for 1 QALY is considered to be 2–3 times the GDP per capita [38–40]. Although the WTP for endoscopic spinal surgery in the ROK has not yet been set, the cost of $5,241 per 1 QALY gained may not be excessive considering the productivity of the inhabitants of the ROK. Therefore, we suggest a realistic reimbursement system for the supplies used during FELD to facilitate a fair suggestion of surgical options to patients. Moreover, FELD could reduce the duration of hospital stay and sick leave, which would thereby also diminish indirect costs [2, 3, 8, 12, 17, 18, 24]. Although FELD was performed under general anesthesia in the current study, FELD could be effectively performed under local or regional anesthesia. Using local or regional anesthesia may enhance the safety and cost-effectiveness of FELD by reducing the costs of anesthesia, the time for recovery from anesthesia, and possibly the hospital stay [45–47].

This advantage would be attractive for countries with high medical costs of hospital stays and sick leave. Although an exact calculation could not be made based on the current data, the same amount ($1,459) of reimbursement for the cost of supplies as in Japan would be a minimal reasonable suggestion.

## Limitations

This study has several limitations. First, it presents results from a small number of patients without control group, which were not sufficient to fully represent the cost-utility of FELD. In addition, this study analyzed only direct medical costs, although indirect medical costs may increase medical expenditures as a whole. However, the main purpose of this study was to show the relevance of the costs incurred at hospitals for FELD. In this regard, we used homogeneous data from patients who underwent the same surgical procedure and were managed with the same clinical pathway in a single non-profit tertiary referral hospital. Although the number of patients was small, the surgical technique, medical costs, and postoperative management were homogeneous. In addition, no additional uninsured surgical materials such as a hemostatic agent or anti-adhesive agent were used, except for the electrode for FELD. We acknowledge that the small number of patients was the major weakness of this study, but the homogeneity may compensate for this limitation. Second, spine surgery is performed at a wide range of hospitals, and the direct costs may be variable across hospitals and countries [8]. The current example of direct medical costs may not be representative, and it limits the generalizability of the findings. If the current suggestion for a practical reimbursement is accepted, data from multiple types of hospitals must be collected to set a realistic reimbursement amount. Last, but not least, FELD was not compared to standard open discectomy, which would have made it possible to truly determine the utility of FELD. However, open discectomies were performed at several hospitals, and the direct medical costs such as hospital stay, room charges, magnetic resonance imaging, and uninsured surgical materials were variable. Therefore, the need to preserve homogeneity meant that open discectomy could not be compared with FELD in this study. Nonetheless, the current study presents an example of cost per QALY after endoscopic spinal surgery, and this approach would be helpful for supporting claims for realistic reimbursements for supplies used for the benefit of patients. A common surgical pathway in care of patients and using surgical materials are required to incorporate FELD as one of appropriate surgical options.

## Conclusion

This study showed that the cost per QALY of FELD was far lower than the GDP per capita of the ROK. For the sake of patients, a fair range of surgical options should be provided, and it is time to consider a practical reimbursement for FELD supplies.

## Supporting information

**S1 Data.**
(XLS)

## Acknowledgments

I would like to express my special thanks to Dr. Kyung-Chul Choi, who shared data and opinions for this paper. I also appreciate the kind assistance of the Korean Spinal Neurosurgery Society for English editing service.

## Author Contributions

**Conceptualization:** Chi Heon Kim, Yunhee Choi, Chang-Hyun Lee, Sung Bae Park, Keewon Kim, Sun Gun Chung.

**Data curation:** Chi Heon Kim, Yunhee Choi, Keewon Kim.

**Formal analysis:** Chi Heon Kim, Yunhee Choi.

**Funding acquisition:** Chi Heon Kim, Chun Kee Chung, Keewon Kim, Sun Gun Chung.

**Investigation:** Chi Heon Kim, Seung Heon Yang.

**Methodology:** Chi Heon Kim, Yunhee Choi.

**Project administration:** Chi Heon Kim.

**Resources:** Chi Heon Kim, Chun Kee Chung.

**Software:** Chi Heon Kim, Yunhee Choi.

**Supervision:** Chun Kee Chung, Sun Gun Chung.

**Validation:** Yunhee Choi.

**Writing – original draft:** Chi Heon Kim, Yunhee Choi, Chun Kee Chung, Seung Heon Yang, Chang-Hyun Lee, Sung Bae Park, Keewon Kim, Sun Gun Chung.

**Writing – review & editing:** Chi Heon Kim, Yunhee Choi, Chun Kee Chung, Seung Heon Yang, Chang-Hyun Lee, Sung Bae Park.

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
