## [Decision Letter · Decision Letter 0]

8 Apr 2022

PONE-D-21-21192Cost-utility analysis of endoscopic lumbar discectomy following a uniform clinical pathway in the Korean national health insurance systemPLOS ONE

Dear Dr. Chung,

Thank you for submitting your manuscript to PLOS ONE. After careful consideration, we feel that it has merit but does not fully meet PLOS ONE’s publication criteria as it currently stands. Therefore, we invite you to submit a revised version of the manuscript that addresses the points raised during the review process.

We look forward to receiving your revised manuscript.

Kind regards,

Jianhong Zhou

Associate Editor

PLOS ONE

Journal Requirements:

3. Thank you for stating the following in the Competing Interests section: "The first author (CHK) is a consultant of RIWOspine GmBH. All the authors declare that they have no conflicts of interest concerning the materials/methods used in this study or the findings described in this paper."

Reviewers' comments:

Reviewer's Responses to Questions

**Comments to the Author**

1. Is the manuscript technically sound, and do the data support the conclusions?

Reviewer #1: Partly

Reviewer #2: Yes

Reviewer #3: Yes

2. Has the statistical analysis been performed appropriately and rigorously? 

Reviewer #1: No

Reviewer #2: Yes

Reviewer #3: Yes

3. Have the authors made all data underlying the findings in their manuscript fully available?

Reviewer #1: Yes

Reviewer #2: Yes

Reviewer #3: No

4. Is the manuscript presented in an intelligible fashion and written in standard English?

Reviewer #1: Yes

Reviewer #2: Yes

Reviewer #3: Yes

5. Review Comments to the Author

Reviewer #1: This paper suffers from a lack of sample size. There appears to be some attempt of a design in the data analysis section on pages 11 and 12. However, these considerations are not clearly written and the justification for 28 subjects is not evident.

The pre post analysis with the generalized mixed model approach is reasonable. However, again, the sample size does not appear adequate and there should have been some consideration of a comparison to the standard open microdiscectomy procedure for cost comparisons. The study is, at best, descriptive. A propensity matching procedure with adequate statistical power should have been attempted to truly determine the utility of the FELD procedure.

Reviewer #2: Nice paper in a gray error of the literature

- Sample size is a bit small

Introduction is well written

The outcomes of open discectomy and endoscopic discectomy are similar, but physicians should inform

2 patients about all possible options and respect their choices cite Gadjradj PS, Rubinstein SM, Peul WC, Depauw PR, Vleggeert-Lankamp CL, Seiger A, van Susante JL, de Boer MR, van Tulder MW, Harhangi BS. Full endoscopic versus open discectomy for sciatica: randomised controlled non-inferiority trial. BMJ. 2022 Feb 21;376:e065846. doi: 10.1136/bmj-2021-065846. PMID: 35190388.

Economic acceptability is usually judged by willingness to pay (WTP), but WTP for FELD has not yet

10 been determined it has now from the Netherlands

Gadjradj PS, Broulikova HM, van Dongen JM, Rubinstein SM, Depauw PR, Vleggeert C, Seiger A, Peul WC, van Susante JL, van Tulder MW, Harhangi BS. Cost-effectiveness of full endoscopic versus open discectomy for sciatica. Br J Sports Med. 2022 Feb 20:bjsports-2021-104808. doi: 10.1136/bjsports-2021-104808. Epub ahead of print. PMID: 35185010.

Please explain perc adhesiolysis: did all consertvative care patients receieve it? Did all surgery patients receive it prior?

Rephrase clinical pathyway

Please explain the rationale for performing surgery under general anesthesia. I know some do this, but why do the authors do this?

Was TELD not even attempted at L5-S1?

Describe when the clinical parameters were measured.

please explain what this electrode is used for.

What about the costs for reoperations?

The ODI is usually presented as a range from 0 to 100

Please create a table in which the costs are broken down: surgery, primary care, medication etc.

Discuss the recently published trial by Gadjradj et al in Br J Sports Med.

Reviewer #3: Dear authors,

I congratulate your efforts for a detailed cost-utility analysis of the FELD procedure to be made available as a choice of surgery to the people ROK based on the QALY gained by the procedure objectively thereby improving the medical services provided to the affected population.

I have a few concerns that needs clarification before this manuscript could be considered for publication.

Rather than utilizing the cost of the electrode alone in the reimbursement calculation you can give an objective model where the hospital offering the tratement need not bear the brunt of the choice of the patient due to the financial constraints involved in the procedure like installation of FELD system, consumables etc., So that a practical statement can be made to the NHIS from this analysis for the recommended method and means of calculating the appropriate reimbursement so that the procedure is beneficial both to the service provider and the patient involved in it.

If needed a comparison can be made between the open discectomy group and the FELD group to establish the superiority of the treatment method over the conventional gold standard procedure currently in use.

6. PLOS authors have the option to publish the peer review history of their article (what does this mean?). If published, this will include your full peer review and any attached files.

Reviewer #1: No

Reviewer #2: **Yes: **Pravesh Shankar Gadjradj

Reviewer #3: **Yes: **Dr Sathish Muthu

---

## [Author Response · Author response to Decision Letter 0]

21 May 2022

Author’s revision letter

Reviewer #1: This paper suffers from a lack of sample size. There appears to be some attempt of a design in the data analysis section on pages 11 and 12. However, these considerations are not clearly written and the justification for 28 subjects is not evident.

Answer. 

I sincerely appreciate your critical comment. This study was a subgroup analysis of prospectively collected data, and the purpose of this study was to analyze the cost-utility of full-endoscopic lumbar discectomy. The number of patients was not pre-determined, and all patients with 2-year follow-up data were included. Although the number of patients was small, all patients underwent the same clinical pathway, surgical technique, and hospital costs incurred during admission. The operations were performed at a university hospital, and no additional uninsured surgical materials, such as a hemostatic agent or anti-adhesive agent, were used, except for the electrode for full endoscopic lumbar discectomy (FELD). This homogeneity of the data may compensate for the weakness of the small number of patients. Nonetheless, the small number of patients was the critical weakness of this study, and this limitation was clarified in the revised manuscript. 

I really appreciate your critical comment. This study was a subgroup analysis of prospectively collected data and the purpose of this study was to analyze cost-utility of full-endoscopic lumbar discectomy. The number of patients were not pre-determined and all patients with 2-year follow-up data were included. Although the number of patients were small, all patients underwent the same clinical pathway, surgical technique and hospital cost incurred during admission. The surgery was performed in a University Hospital, and any additional un-insured surgical materials such as hemostatic agent, anti-adhesive agent were not used, except for the electrode for full endoscopic lumbar discectomy (FELD). This homogeneity of data may make up the weakness of small number of patients. Nonetheless, small number of patients was the critical weakness of this study, and the limitation was clarified in a revised manuscript. 

Page 17, line 69

This study has several limitations. First, it presents results from a small number of patients, which were not sufficient to fully represent the cost-utility of FELD. In addition, this study analyzed only direct medical costs, although indirect medical costs may increase medical expenditures as a whole. However, the main purpose of this study was to show the relevance of the costs incurred at hospitals for FELD. In this regard, we used homogeneous data from patients who underwent the same surgical procedure and were managed with the same clinical pathway in a single tertiary referral hospital. Although the number of patients was small, the surgical technique, medical costs, and postoperative management were homogeneous. In addition, the surgery was performed at a university hospital, and no additional uninsured surgical materials such as a hemostatic agent or anti-adhesive agent were used, except for the electrode for FELD. We acknowledge that the small number of patients was the major weakness of this study, but the homogeneity may compensate for this limitation.

First, results from small number of patients were not sufficient to represent cost-utility of FELD. In addition, this study analyzed only direct medical costs, although indirect medical costs may increase medical expenditures as a whole. However, the main purpose of this study was to show the relevance of the costs incurred at hospitals for FELD. In this regard, we used homogeneous data from patients who underwent the same surgical procedure and were managed with the same clinical pathway in a single tertiary referral hospital. Although the number of patients were small, the surgical technique, medical cost and postoperative management were homogeneous. In addition, the surgery was done in a University Hospital, and any additional un-insured surgical materials such as hemostatic agent, anti-adhesive agent were not used, except for the electrode for FELD. We acknowledged that the small number of patients was the major weakness of this study, but the homogeneity may make up the weakness of small number of patients.

The pre post analysis with the generalized mixed model approach is reasonable. However, again, the sample size does not appear adequate and there should have been some consideration of a comparison to the standard open microdiscectomy procedure for cost comparisons. The study is, at best, descriptive. A propensity matching procedure with adequate statistical power should have been attempted to truly determine the utility of the FELD procedure.

Answer. 

I totally agree with your opinion. A comparative study with a sufficient number of patients should have been done to effectively show the cost-effectiveness of FELD. However, the open surgeries discectomies were performed in at other several other hospitals and the direct medical costs were variable among hospitals. The purpose of this study was to show a the cost-utility of FELD following a uniform clinical pathway in the Korean national health insurance system and to present an the actual costs of FELD for to support a realistic reimbursement. In this regard, we included homogeneous data of from a university hospital that which strictly follows the regulations of the national health insurance system and minimizes the use of un-insured surgical materials. Nonetheless, a comparative study should have been attempted to truly determine the utility of the FELD procedure. We have acknowledged the this limitation and described it in the limitations section. 

Page 17, line 213 

Last, but not least, FELD was not compared to standard open discectomy, which would have made it possible to truly determine the utility of FELD. However, open discectomies were performed at several hospitals, and the direct medical costs such as hospital stay, room charges, magnetic resonance imaging, and uninsured surgical materials were variable. Therefore, the need to preserve homogeneity meant that open discectomy could not be compared with FELD in this study. Last, but not least, FELD was not compared to a standard open discectomy. The cost-effectiveness should have been compared with a standard open discectomy to truly determine the utility of FELD. However, the open discectomies were performed at several hospital and direct medical cost such as hospital stay, room charge, magnetic resonance imaging, and un-insured surgical materials were variable. Therefore, considering homogeneity, open discectomy could not be compared with FELD in this study.

 

Reviewer #2: Nice paper in a gray error of the literature

- Sample size is a bit small

Answer. 

I sincerely appreciate your critical comment. This study was a subgroup analysis of prospectively collected data, and the purpose of this study was to analyze the cost-utility of full-endoscopic lumbar discectomy. The number of patients was not pre-determined, and all patients with 2-year follow-up data were included. Although the number of patients was small, all patients underwent the same clinical pathway, surgical technique, and hospital costs incurred during admission. The operations were performed at a university hospital, and no additional uninsured surgical materials, such as a hemostatic agent or anti-adhesive agent, were used, except for the electrode for full endoscopic lumbar discectomy (FELD). This homogeneity of the data may compensate for the weakness of the small number of patients. Nonetheless, the small number of patients was the critical weakness of this study, and this limitation was clarified in the revised manuscript. 

I really appreciate your critical comment. This study was a subgroup analysis of prospectively collected data and the purpose of this study was to analyze cost-utility of full-endoscopic lumbar discectomy. The number of patients were not pre-determined and all patients with 2-year follow-up data were included. Although the number of patients were not small, all patients underwent the same clinical pathway, surgical technique and hospital cost incurred during admission. The surgery was performed in a University Hospital, and any additional un-insured surgical materials such as hemostatic agent, anti-adhesive agent were not used, except for the electrode for full endoscopicl lumbar discectomy (FELD). This homogeneity of data may make up the weakness of small number of patients. Nonetheless, small number of patients was the critical weakness of this study, and the limitation was clarified in a revised manuscript. 

Page 17, line 69

First, it presents results from a small number of patients, which were not sufficient to fully represent the cost-utility of FELD. In addition, this study analyzed only direct medical costs, although indirect medical costs may increase medical expenditures as a whole. However, the main purpose of this study was to show the relevance of the costs incurred at hospitals for FELD. In this regard, we used homogeneous data from patients who underwent the same surgical procedure and were managed with the same clinical pathway in a single tertiary referral hospital. Although the number of patients was small, the surgical technique, medical costs, and postoperative management were homogeneous. In addition, the surgery was performed at a university hospital, and no additional uninsured surgical materials such as a hemostatic agent or anti-adhesive agent were used, except for the electrode for FELD. We acknowledge that the small number of patients was the major weakness of this study, but the homogeneity may compensate for this limitation.First, results from small number of patients were not sufficient to represent cost-utility of FELD. In addition, this study analyzed only direct medical costs, although indirect medical costs may increase medical expenditures as a whole. However, the main purpose of this study was to show the relevance of the costs incurred at hospitals for FELD. In this regard, we used homogeneous data from patients who underwent the same surgical procedure and were managed with the same clinical pathway in a single tertiary referral hospital. Although the number of patients were small, the surgical technique, medical cost and postoperative management were homogeneous. In addition, the surgery was done in a University Hospital, and any additional un-insured surgical materials such as hemostatic agent, anti-adhesive agent were not used, except for the electrode for FELD. We acknowledged that the small number of patients was the major weakness of this study, but the homogeneity may make up the weakness of small number of patients.

Introduction is well written

The outcomes of open discectomy and endoscopic discectomy are similar, but physicians should inform to patients about all possible options and respect their choices cite Gadjradj PS, Rubinstein SM, Peul WC, Depauw PR, Vleggeert-Lankamp CL, Seiger A, van Susante JL, de Boer MR, van Tulder MW, Harhangi BS. Full endoscopic versus open discectomy for sciatica: randomised controlled non-inferiority trial. BMJ. 2022 Feb 21;376:e065846. doi: 10.1136/bmj-2021-065846. PMID: 35190388.

Answer. 

Thank you very much for the recommendation. I We included and discussed the that valuable article in the present study. 

Economic acceptability is usually judged by willingness to pay (WTP), but WTP for FELD has not yet been determined it has now from the Netherlands

Gadjradj PS, Broulikova HM, van Dongen JM, Rubinstein SM, Depauw PR, Vleggeert C, Seiger A, Peul WC, van Susante JL, van Tulder MW, Harhangi BS. Cost-effectiveness of full endoscopic versus open discectomy for sciatica. Br J Sports Med. 2022 Feb 20:bjsports-2021-104808. doi: 10.1136/bjsports-2021-104808. Epub ahead of print. PMID: 35185010.

Answer. 

I appreciate your comment and recommendation of those articles. As the recommended study showed, a comparative cost-effective analysis will be necessary to include endoscopic lumbar discectomy in the package of health insurance. Although this study was not designed like the recommended article, the recommended articles will waeres included be as an important references in this study. 

Page 5, line 19

A recent randomized controlled multi-center trial by Gadjradj et al. showed better clinical outcomes after FELD than after open discectomy, and the cost of surgery was lower with FELD than with open discectomy.[4,16] Economic acceptability is usually judged by willingness to pay (WTP), and those studies showed the WTP for FELD.[4,16] The results enabled FELD to be included in the health insurance package of the Netherlands.A recent randomized controlled multi-center trial by Gadjradj et al. showed better clinical outcome after FELD than after open discectomy and the cost of surgery was lower with FELD than open discectomy.[4,17] Economic acceptability is usually judged by willingness to pay (WTP), and those studies showed WTP) for FELD.[4,17] The results enabled FELD to be included in the health insurance package of Netherland.[4,17]

The choice among surgical options would be fairer if the actual costs of those supplies were reimbursed, but WTP for FELD has not yet been determined in the ROK.[8-10]Page 6, line 6

The choice among surgical options would be fairer if the actual costs of those supplies were reimbursed, but WTP for FELD has not yet been determined in ROK.[7-9]

Page 16, line 1429

Recently, Gadjradj et al. showed better leg pain and QALYs after endoscopic lumbar discectomy than after open discectomy.[4,16] through a randomized controlled trial. This study showed that the WTP for endoscopic lumbar discectomy will be similar or higher than the WTP of 6112 Euros ($6612) in the Netherlands.[4,16] Those results prompted the Dutch Ministry of Health to include endoscopic lumbar discectomy in the basic health insurance package.[4,16]Recently, Gadjradj et al. showed better leg pain and QALYs after endoscopic lumbar discectomy than open discectomy.[4,17] through a randomized controlled trail. This study showed that WTP of endoscopic lumbar discectomy will be similar or higher than 6112 Euro ($6612) in Netherland.[4,17] Those results enabled the Dutch Ministry of Health include endoscopic lumbar discectomy in the basic health insurance package.[4,17]

Please explain perc adhesiolysis: did all consertvative care patients receieve it? Did all surgery patients receive it prior?

Answer. 

I appreciate the comment. As you pointed out, adhesiolysis is not a routine procedure and choice of non-surgical treatment was at the discretion of doctors. The sentence may confuse readers, and so it was deleted. 

Rephrase clinical pathyway

Answer. 

Thank you for giving us a chance to introduce the clinical pathway. It was rephrased as follows. 

Page 8, line 235

Clinical pathway of endoscopic lumbar discectomy

All patients were managed with the same protocol and orders (clinical pathway), which were developed by the clinicians.[28] When the decision was made to perform surgery, basic preoperative check-ups such as blood tests, electrocardiography, chest X-rays, and spinal X-rays were performed at the outpatient clinic. On day 1 (preoperative day 1), the patient was admitted in the afternoon for a preoperative check-up by an anesthesiologist and other physicians, and provided informed consent for surgery. On day 2 (the day of surgery), the patient was transferred to the operation room after a sensitivity test for antibiotics (first-generation cephalosporin) and underwent surgery under general anesthesia. A Foley catheter was not inserted. After the operation, the patient was moved to the post-anesthesia care unit and was returned to the general ward after fully awakening, as confirmed by an anesthesiologist. The patient was encouraged to stand up and walk around the hall of the general ward with a walking aid within 1 hour of returning to the general ward. Pain was controlled with oral acetaminophen and a regular diet was allowed in the evening after confirmation of bowel sound or passing flatussounds. Postoperative magnetic resonance imaging (MRI) was taken during the evening or night to check the results of discectomy. On day 3 (postoperative day 1), the patient was discharged home in the afternoon after confirmation of independent ambulation, improvements of preoperative symptoms/signs and the absence of any postoperative complication such as a neurological problem. After discharge, a lumbar-supporting brace was not applied, and office work was encouraged within 1 month after surgery. However, strenuous activities such as sports, vigorous leisure activities, and weightlifting were not allowed until 3 months after surgery. The patient was scheduled to visit the outpatient clinic at postoperative 1, 3, 6, and 12 months and yearly thereafter. If pain recurred, it was managed with drugs and/or epidural injections on an outpatient basis. If the pain was unbearable or those measures were not effective for 2 months, MRI was recommended to check for the possibility of recurrence. 

Clinical pathway of endoscopic lumbar discectomy

All patients were managed according towith the same protocol and orders (clinical pathway), which were developed by the clinicians.[30] When a surgery was decided, basic preoperative check-ups such as blood test, electrocardiography, chest X-ray and spinal X rays were performed at out-patients’ clinic. On day I (preoperative day 1), TtheIn this pathway, a patient is admitted in the afternoon one day before surgery for preoperative check-up by ato meet the anesthesiologist and other physicians, and for the operation and to provide informed consent for surgery. On day 2 (On the day of surgery), the patient is transferred to the operation room after a sensitivity test for antibiotics (first-generation cephalosporin) and undergoes surgery under general anesthesia. A Foley catheter is not inserted. After the operation, the patient is moved to the post-anesthesia care unit and is returned to the general ward after fully awakening, as confirmed by an anesthesiologist. The patient is encouraged to stand up and walk around the hall of the general ward with an walking aid within 1 hour of returning to general ward. Pain is controlled with oral acetaminophen and a regular diet is allowed in the evening after listening of bowel sound. A postoperative magnetic resonance imaging (MRI) is taken during the evening or night to check discectomy. On day 3 (postoperative day 1), The next day, the patient is discharged home in the afternoon after confirmation of independent ambulation, improvement of preoperative symptom/sign and the absence of any postoperative complication such as neurological problem. After discharge, Pain is controlled with oral acetaminophen. Nlumbar-supporting brace is not applied and office work is encouraged within 1 month after surgery. However, strenuous activities such as sports, vigorous leisure activities, and weightlifting are not allowed until 3 months after surgery. The patient is scheduled to visit the outpatient clinic at postoperative 1, 3, 6, and 12 months and yearly thereafter. If pain recurs, it is managed with drugs and/or epidural injections on an outpatient basis. When the pain is unbearable or those measures were not effective for 2 months, MRI is recommended to check recurrence when those measures are not effective for 2 months.. 

Please explain the rationale for performing surgery under general anesthesia. I know some do this, but why do the authors do this?

Answer. 

Your comment isThank you for this relevant comment. Many surgeons do perform endoscopic surgery under local or regional anesthesia. I started to learn endoscopic lumbar discectomy by from a surgeons who do performed it under general anesthesia (Dr. Sebastian Ruetten in St. Anna Hospital, Germany). I also know that many surgeons prefer local or regional anesthesia to be able to receive get feedback from patients. Moreover, postoperative recovery is better with local or regional anesthesia than general anesthesia, and hospital costs could be reduced by local or regional anesthesia. However, it was not easy to change my routine practice, and I use intraoperative neuromonitoring and the clinical pathway for those purposes. I clarified it this point in the discussion section as follows. 

Page 186, line 1528

Although FELD was performed under general anesthesia in the current study, FEED could be effectively performed under local or regional anesthesia. Using local or regional anesthesia may enhance the safety and cost-effectiveness of FELD by reducing the costs of anesthesia, the time for recovery from anesthesia, and possibly the hospital stay.[45-47]Although FELD was performed under general anesthesia in the current study, FEED could be effectively performed under local or regional anesthesia. It may enhance safety and cost-effectiveness of FELD by reducing cost of anesthesia, the time for recovery from anesthesia, and possibly hospital stay.[49,50]

Was TELD not even attempted at L5-S1?

Answer. 

There is evidence of equal efficacy of the transforaminal and interlaminar approach at L5-S1 and the selection of the approach was at the surgeon’s discretion of surgeon. I prefer the interlaminar approach for L5-S1 level, and a transforaminal approach is was not attempted. 

Describe when the clinical parameters were measured.

Answer. 

Thank you for the comment. The schedule was described in the revised manuscript. 

Page 98, line 1324

All participants were asked to complete patient-reported outcome questionnaires before the operation and at postoperative 1, 3, 6, 12, and 24 monthsAll participants were asked to complete patient-reported outcome questionnaires before operation and at postoperative 1, 3, 6, 12, and 24 months.

please explain what this electrode is used for.

Answer. 

Thank you for the comment. I We described the usage of the electrode in the introduction section. 

Page 6, line 34 

The electrode is an especially essential tool in FELD. It is specifically designed for the surgeon to navigate the tip of the electrode into the spinal canal through a spinal endoscope, and it is used for coagulation, shrinkage of tissue, and dissection.The electrode is an especially essential tool in FELD. It is specifically designed to navigate the tip of electrode into the spinal canal through spinal endoscope and it is used for coagulation, shrinkage of tissue and dissection

What about the costs for reoperations?

Answer. 

In the ROK, the cost of a reoperation is the same as the cost of the first operation. The cost of the operation is set by according to the procedure of surgery, and the difficulty in of revision surgery is not considered. 

The ODI is usually presented as a range from 0 to 100

Answer. 

I appreciate your comment and the score was presented as a range from 0 to 100 in a the revised manuscript. Please see the changes throughout the manuscript.

Please create a table in which the costs are broken down: surgery, primary care, medication etc.

Answer. 

Thank you for the advicesuggestion. The costs were broken down into the costs of surgery, room, electrode, and general care. Among these costs, the costs of the room and magnetic resonance images imaging were variable, and it they was were described like as followingfollows. 

Table 3. Itemized costs at postoperative 1 month

Itemized cost Cost ($)

Surgery 625

Electrode 700

Anesthesia 155

Room (mean ± SD) 95±97

Single* 3

Twin* 10

Shared* 15

MRI (mean ± SD) 867±311

Pre-op* 7

Post-op* 28

Meal 16

Total cost, mean (95% CI)† 3,255 (2,974, 3,536)

*Number of patients.

SD, standard deviation; MRI, magnetic resonance imaging; Pre-op, preoperative; Post-op, postoperative.

†Total included the costs of itemized costs and other non-itemized costs such as general care and nursing. 

Table 3. Cost at postoperative 1 month

Details Cost ($)

Surgery 625

Electrode 700

Anesthesia 155

Room (mean ± SD) 95±97

Single* 3

Twin* 10

Shared* 15

MRI (mean ± SD) 867±311

pre op* 7

post op* 28

Meal 16

Total cost mean (95% confidence interval)† 3,255 (2,974, 3,536)

*Number of patients

†Total included costs of details and other costs such as general care, nursing, and medication. 

Discuss the recently published trial by Gadjradj et al in Br J Sports Med.

Answer. 

Thank you for recommending these relevant articles. ; Those articlesthey were discussed below as follows. 

Page 5, line 19

A recent randomized controlled multi-center trial by Gadjradj et al. showed better clinical outcomes after FELD than after open discectomy, and the cost of surgery was lower with FELD than with open discectomy.[4,16] Economic acceptability is usually judged by willingness to pay (WTP), and those studies showed the WTP for FELD.[4,16] The results enabled FELD to be included in the health insurance package of the Netherlands.[4,16]A recent randomized controlled multi-center trial by Gadjradj et al. showed better clinical outcome after FELD than after open discectomy and the cost of surgery was lower with FELD than open discectomy.[4,17] Economic acceptability is usually judged by willingness to pay (WTP), and those studies showed WTP) for FELD.[4,17] The results enabled FELD to be included in the health insurance package of Netherland.[4,17]

Page 6, line 6

The choice among surgical options would be fairer if the actual costs of those supplies were reimbursed, but WTP for FELD has not yet been determined in the ROK.[8-10]The choice among surgical options would be fairer if the actual costs of those supplies were reimbursed, but WTP for FELD has not yet been determined in ROK.[7-9]

Page 16, line 1529

Recently, Gadjradj et al. showed better leg pain and QALYs after endoscopic lumbar discectomy than after open discectomy.[4,16] through a randomized controlled trial. This study showed that the WTP for endoscopic lumbar discectomy will be similar or higher than the WTP of 6112 Euros ($6612) in the Netherlands.[4,16] Those results prompted the Dutch Ministry of Health to include endoscopic lumbar discectomy in the basic health insurance package.[4,16]Recently, Gadjradj et al. showed better leg pain and QALYs after endoscopic lumbar discectomy than open discectomy.[4,17] through a randomized controlled trail. This study showed that WTP of endoscopic lumbar discectomy will be similar or higher than 6112 Euro ($6612) in Netherland.[4,17] Those results enabled the Dutch Ministry of Health include endoscopic lumbar discectomy in the basic health insurance package.[4,17]

 

Reviewer #3: Dear authors,

I congratulate your efforts for a detailed cost-utility analysis of the FELD procedure to be made available as a choice of surgery to the people ROK based on the QALY gained by the procedure objectively thereby improving the medical services provided to the affected population.

I have a few concerns that needs clarification before this manuscript could be considered for publication.

Rather than utilizing the cost of the electrode alone in the reimbursement calculation you can give an objective model where the hospital offering the tratement need not bear the brunt of the choice of the patient due to the financial constraints involved in the procedure like installation of FELD system, consumables etc., So that a practical statement can be made to the NHIS from this analysis for the recommended method and means of calculating the appropriate reimbursement so that the procedure is beneficial both to the service provider and the patient involved in it.

Answer. 

I totally agree with your opinion, and hope to adopt the system in National Health Insurance system (NHIS). However, the goal of the Korean NHIS is to provide the same level of medical service to all patients irrespective of their wealth or insurance status. So far, not all hospital costs are covered by NHIS and un-insured costs are onare borne by patients. The government has been taking steps to gradually includes all possible medical services under the control of NHIS, such as magnetic resonance image imaging, and has attempted to control the inflation of medical costs by using the regulation ofby the NHIS regulations. Therefore, your model may not be working precisely suit under the current NHIS of the ROK. 

If needed, a comparison can be made between the open discectomy group and the FELD group to establish the superiority of the treatment method over the conventional gold standard procedure currently in use.

Answer. 

Answer. 

I totally agree with your opinion. A comparative study with a sufficient number of patients should have been done to effectively show the cost-effectiveness of FELD. However, the open discectomies were performed at several other hospitals and the direct medical costs were variable among hospitals. The purpose of this study was to show the cost-utility of FELD following a uniform clinical pathway in the Korean national health insurance system and to present the actual costs of FELD to support a realistic reimbursement. In this regard, we included homogeneous data from a university hospital that strictly follows the regulations of the national health insurance system and minimizes the use of un-insured surgical materials. Nonetheless, a comparative study should have been attempted to truly determine the utility of the FELD procedure. We have acknowledged this limitation and described it in the limitations section. 

I totally agree with your opinion. A comparative study with a sufficient number of patients should have been done to effectively show cost-effectiveness of FELD. However, the open surgeries were performed in other several hospitals and the direct medical costs were variable among hospitals. The purpose of this study was to show a cost-utility of FELD following a uniform clinical pathway in the Korean national health insurance system and to present an actual cost of FELD for a realistic reimbursement. In this regard, we included homogeneous data of university hospital which strictly follow regulation of national health insurance system and minimize the use of un-insured surgical materials. Nonetheless, a comparative study should have been attempted to truly determine the utility of the FELD procedure. We acknowledged the limitation and described it in limitation section. 

Page 17, line 8 

This study has several limitations. First, it presents results from a small number of patients, which were not sufficient to fully represent the cost-utility of FELD. In addition, this study analyzed only direct medical costs, although indirect medical costs may increase medical expenditures as a whole. However, the main purpose of this study was to show the relevance of the costs incurred at hospitals for FELD. In this regard, we used homogeneous data from patients who underwent the same surgical procedure and were managed with the same clinical pathway in a single tertiary referral hospital. Although the number of patients was small, the surgical technique, medical costs, and postoperative management were homogeneous. In addition, the surgery was performed at a university hospital, and no additional uninsured surgical materials such as a hemostatic agent or anti-adhesive agent were used, except for the electrode for FELD. We acknowledge that the small number of patients was the major weakness of this study, but the homogeneity may compensate for this limitation.

Second, spine surgery is performed at a wide range of hospitals, and the direct costs may be variable across hospitals and countries.[8] The current example of direct medical costs may not be representative, and it limits the generalizability of the findings. If the current suggestion for a practical reimbursement is accepted, data from multiple types of hospitals must be collected to set a realistic reimbursement amount. Last, but not least, FELD was not compared to standard open discectomy, which would have made it possible to truly determine the utility of FELD. However, open discectomies were performed at several hospitals, and the direct medical costs such as hospital stay, room charges, magnetic resonance imaging, and uninsured surgical materials were variable. Therefore, the need to preserve homogeneity meant that open discectomy could not be compared with FELD in this study. Limitations

This study has several limitations. First, results from small number of patients were not sufficient to represent cost-utility of FELD. In addition, this study analyzed only direct medical costs, although indirect medical costs may increase medical expenditures as a whole. However, the main purpose of this study was to show the relevance of the costs incurred at hospitals for FELD. In this regard, we used homogeneous data from patients who underwent the same surgical procedure and were managed with the same clinical pathway in a single tertiary referral hospital. Although the number of patients were small, the surgical technique, medical cost and postoperative management were homogeneous. In addition, the surgery was done in a University Hospital, and any additional un-insured surgical materials such as hemostatic agent, anti-adhesive agent were not used, except for the electrode for FELD. We acknowledged that the small number of patients was the major weakness of this study, but the homogeneity may make up the weakness of small number of patients.

First, the small number of patients was the major weakness of this study. In addition, this study analyzed only direct medical costs, although indirect medical costs may increase medical expenditures as a whole. However, the main purpose of this study was to show the relevance of the costs incurred at hospitals for FELD. In this regardMoreover, we used homogeneous data from patients who underwent the same surgical procedure and were managed with followed the same clinical pathway at a single tertiary referral hospital. Any un-insured surgical materials such as hemostatic agent, anti-adhesive agent were not used, except for the electrode for FELD. We acknowledged that the number of patients was the major weakness of this study, but the homogeneity may make up the weakness of small number of patients. Second, spine surgery is performed at a wide range of hospitals, and the direct costs may be variablenot be the same.[7] The current example of direct medical costs may not be representative, and it which limits the generalizability of the findings. If the current suggestion for a practical reimbursement is accepted, data from multiple types of hospitals must be collected to set a realistic reimbursement amount. Last, but not least, FELD was not compared to a standard open discectomy. The cost-effectiveness should have been presented compared towihtwith a standard open discectomy to truly determine the utility of FELD. However, the open discectomies were performed at several hospital and direct medical cost such as hospital stay, room charge, magnetic resonance imaging, and un-insured surgical materials were variable. Therefore, considering homogeneity, open discectomy could not be compared with FELD in this study. Nonetheless, the current study presents an example of cost per QALY after endoscopic spinal surgery, and this approach would be helpful for supporting claims for realistic levant reimbursements for supplies used for the benefit of patients.

---

## [Decision Letter · Decision Letter 1]

13 Jun 2022

PONE-D-21-21192R1Cost-utility analysis of endoscopic lumbar discectomy following a uniform clinical pathway in the Korean national health insurance systemPLOS ONE

Dear Dr. Chung,

Thank you for submitting your manuscript to PLOS ONE. After careful consideration, we feel that it has merit but does not fully meet PLOS ONE’s publication criteria as it currently stands. Therefore, we invite you to submit a revised version of the manuscript that addresses the points raised during the review process.

We look forward to receiving your revised manuscript.

Kind regards,

Sathish Muthu

Guest Editor

PLOS ONE

Additional Editor Comments (if provided):

Reviewer 3

Dear Authors

I find that you have addressed a few of the queries raised in the previous round of review. Having looking into your intervention cohort, i find a significant number of patient also underwent open discectomy with follow-up data as given in Figure file. I urge the authors to include them into the analysis and make the study robust since the current analysis suffers from low number of patients included for analysis and lack of comparison group. Moreover, for inclusion into any health system a comparative analysis of the existing standard to the newer technique needs to be presented for acceptance

Reviewer 4

I congratulate the authors for the research work. The article tries to address one of the important bottlenecks in adapting - the cost of the procedure. However the article falls short of providing good scientifically relevant information for the readers. The important conclusion of the article is that the cost per QALY of FELD is less than the GDP per capita of ROK. This is more like a recommendation to a health administration rather than a scientific material for publication.

The authors have compared the outcomes of FELD with open discectomy right from the introduction. The article would have been more informative if the cost per QALY of both these were compared. I strongly recommend the authors to include the cost per QALY of the open discectomy as well.

Methods: this study is a subgroup analysis of a previous study. Lines 1 to 9 of page 7 can be removed.

Line 16 page 9: Purpose of the study should be at the end of introduction and should be clear.

Results are well written

Discussion:

What is the additional relevant information provided in this study when compared to the Choi et al or Kim et al study? (reference 8 and 12). this should be included in discussion.

Line 29, page - correct FEED to FELD

General discussion on the method used for calculation of Cost per QALY should be added along with the comparison of other methods available.

Limitations:

lines 17,18 page 17 are getting repeated and can be removed.

lines 26,27 page 17, in the parent study of this study, there are patients who underwent open discectomy. Weren't operated in the same hospital? Comparing them for cost utility will give homogeneity.

Reviewers' comments:

Reviewer's Responses to Questions

**Comments to the Author**

1. If the authors have adequately addressed your comments raised in a previous round of review and you feel that this manuscript is now acceptable for publication, you may indicate that here to bypass the “Comments to the Author” section, enter your conflict of interest statement in the “Confidential to Editor” section, and submit your "Accept" recommendation.

Reviewer #2: All comments have been addressed

Reviewer #3: (No Response)

Reviewer #4: (No Response)

2. Is the manuscript technically sound, and do the data support the conclusions?

Reviewer #2: Yes

Reviewer #3: Partly

Reviewer #4: Yes

3. Has the statistical analysis been performed appropriately and rigorously? 

Reviewer #2: Yes

Reviewer #3: Yes

Reviewer #4: Yes

4. Have the authors made all data underlying the findings in their manuscript fully available?

Reviewer #2: Yes

Reviewer #3: Yes

Reviewer #4: Yes

5. Is the manuscript presented in an intelligible fashion and written in standard English?

Reviewer #2: Yes

Reviewer #3: Yes

Reviewer #4: Yes

6. Review Comments to the Author

Reviewer #2: Thank you for adressing the issues and for a thorough revision. I support publication. Congrats with your work!

Reviewer #3: Dear Authors

I find that you have addressed a few of the queries raised in the previous round of review. Having looking into your intervention cohort, i find a significant number of patient also underwent open discectomy with follow-up data as given in Figure file. I urge the authors to include them into the analysis and make the study robust since the current analysis suffers from low number of patients included for analysis and lack of comparison group. Moreover, for inclusion into any health system a comparative analysis of the existing standard to the newer technique needs to be presented for acceptance.

Reviewer #4: I congratulate the authors for the research work. The article tries to address one of the important bottlenecks in adapting - the cost of the procedure. However the article falls short of providing good scientifically relevant information for the readers. The important conclusion of the article is that the cost per QALY of FELD is less than the GDP per capita of ROK. This is more like a recommendation to a health administration rather than a scientific material for publication.

The authors have compared the outcomes of FELD with open discectomy right from the introduction. The article would have been more informative if the cost per QALY of both these were compared. I strongly recommend the authors to include the cost per QALY of the open discectomy as well.

Methods: this study is a subgroup analysis of a previous study. Lines 1 to 9 of page 7 can be removed.

Line 16 page 9: Purpose of the study should be at the end of introduction and should be clear.

Results are well written

Discussion:

What is the additional relevant information provided in this study when compared to the Choi et al or Kim et al study? (reference 8 and 12). this should be included in discussion.

Line 29, page - correct FEED to FELD

General discussion on the method used for calculation of Cost per QALY should be added along with the comparison of other methods available.

Limitations:

lines 17,18 page 17 are getting repeated and can be removed.

lines 26,27 page 17, in the parent study of this study, there are patients who underwent open discectomy. Weren't operated in the same hospital? Comparing them for cost utility will give homogeneity.

7. PLOS authors have the option to publish the peer review history of their article (what does this mean?). If published, this will include your full peer review and any attached files.

Reviewer #2: **Yes: **Pravesh Shankar Gadjradj

Reviewer #3: **Yes: **Dr Sathish Muthu

Reviewer #4: **Yes: **Girinivasan Chellamuthu

---

## [Author Response · Author response to Decision Letter 1]

21 Apr 2023

Author’s response to reviewers. 

Reviewer #2: Thank you for adressing the issues and for a thorough revision. I support publication. 

Congrats with your work!

Answer. 

I appreciate the valuable comments and those were helpful in improving this paper. Thank you very much. 

Reviewer #3: Dear Authors

I find that you have addressed a few of the queries raised in the previous round of review. Having looking into your intervention cohort, i find a significant number of patient also underwent open discectomy with follow-up data as given in Figure file. I urge the authors to include them into the analysis and make the study robust since the current analysis suffers from low number of patients included for analysis and lack of comparison group. Moreover, for inclusion into any health system a comparative analysis of the existing standard to the newer technique needs to be presented for acceptance.

Answer. 

I highly appreciate your advice and the comparison between endoscopic surgery and the standard open discectomy will make this study robust. Although comparing cost-utility between full endoscopic lumbar discectomy (FELD) and open discectomy may have been the best study design, patients with open discectomy did not follow the clinical pathway and underwent surgeries in other hospitals. To maintain the homogeneity of data, we only analyzed patients with FELD, who underwent surgery in a non-profit tertiary hospital with a uniform clinical pathway. 

During revision of manuscript, we tried to obtain cost data of patietns with open discectomy, but institutional review board did not approve to open sensitive cost data to the other hospital, because this study was a post hoc analysis and information of cost had not been planned from the beginning. Because special single-used surgical instruments are necessary for FELD, we should show the practical economical burden of FELD. Although this study included a small number of patients, the care of patients were standardized using a uniform clinical pathway and the direct cost was minimized in a non-profit tertiary hospital. Therefore, this study may be meaningful in showing the minimal burden of surgical cost and instruments of FELD. The current study may be referenced in countries with national health insruace to embrace economical burden of the hospital and patients by government. Nonetheless, we acknowledged the limitation and clarified it in the limitation section as followed. 

Page 21, line 11

First, it presents results from a small number of patients without control group, which were not sufficient to fully represent the cost-utility of FELD. In addition, this study analyzed only direct medical costs, although indirect medical costs may increase medical expenditures as a whole. However, the main purpose of this study was to show the relevance of the costs incurred at hospitals for FELD. In this regard, we used homogeneous data from patients who underwent the same surgical procedure and were managed with the same clinical pathway in a single non-profit tertiary referral hospital. Although the number of patients was small, the surgical technique, medical costs, and postoperative management were homogeneous. In addition, no additional uninsured surgical materials such as a hemostatic agent or anti-adhesive agent were used, except for the electrode for FELD. We acknowledge that the small number of patients was the major weakness of this study, but the homogeneity may compensate for this limitation.

Reviewer #4: I congratulate the authors for the research work. The article tries to address one of the important bottlenecks in adapting - the cost of the procedure. However the article falls short of providing good scientifically relevant information for the readers. The important conclusion of the article is that the cost per QALY of FELD is less than the GDP per capita of ROK. This is more like a recommendation to a health administration rather than a scientific material for publication.

The authors have compared the outcomes of FELD with open discectomy right from the introduction. The article would have been more informative if the cost per QALY of both these were compared. I strongly recommend the authors to include the cost per QALY of the open discectomy as well.

Answer. 

I highly appreciate your advice and the comparison between endoscopic surgery and the standard open discectomy will make this study robust. The purpose of this study was to show a cost-utility of full endoscopic lumbar discectomy (FELD) and showed an example of assessing cost-utility by using GDP of ROK. Nonetheless of the purpose, the current study may have limitation in not including control group (standard open discectomy). Although comparing cost-utility between FELD and open discectomy may have been the best study design, patients with open discectomy did not follow the clinical pathway and underwent surgeries in other hospitals. To maintain the homogeneity of data, we only analyzed patients with FELD, who underwent surgery in a non-profit tertiary hospital with a uniform clinical pathway. 

During revision of manuscript, we tried to obtain cost data of patietns with open discectomy, but institutional review board did not approve to open sensitive cost data to the other hospital, because this study was a post hoc analysis and information of cost had not been planned from the beginning. Because special single-used surgical instruments are necessary for FELD, we should show the practical economical burden of FELD. Although this study included a small number of patients, the care of patients were standardized using a uniform clinical pathway and the direct cost was minimized in a non-profit tertiary hospital. Therefore, this study may be meaningful in showing the minimal burden of surgical cost and instruments of FELD. The current study may be referenced in countries with national health insruace to embrace economical burden of the hospital and patients by government. Nonetheless, we acknowledged the limitation and clarified it in the limitation section as followed. 

Page 21, line 11

First, it presents results from a small number of patients without control group, which were not sufficient to fully represent the cost-utility of FELD. In addition, this study analyzed only direct medical costs, although indirect medical costs may increase medical expenditures as a whole. However, the main purpose of this study was to show the relevance of the costs incurred at hospitals for FELD. In this regard, we used homogeneous data from patients who underwent the same surgical procedure and were managed with the same clinical pathway in a single non-profit tertiary referral hospital. Although the number of patients was small, the surgical technique, medical costs, and postoperative management were homogeneous. In addition, no additional uninsured surgical materials such as a hemostatic agent or anti-adhesive agent were used, except for the electrode for FELD. We acknowledge that the small number of patients was the major weakness of this study, but the homogeneity may compensate for this limitation.

Methods: this study is a subgroup analysis of a previous study. Lines 1 to 9 of page 7 can be removed.

Line 16 page 9: Purpose of the study should be at the end of introduction and should be clear.

Answer. 

I appreciate your comments. I deleted the paragraph and changed the purpose of this study in the introduction section as follows. 

Page 6, line 6

The choice among surgical options would be fairer if the actual costs of those supplies were reimbursed, but WTP for FELD has not yet been determined in the ROK.[8-10] To address this gap, we provided a cost-utility of FELD and suggested an appropriate cost for reimbursement of the supplies used in FELD.[11]

Results are well written

Answer. 

I appreciate your encouragement. 

Discussion:

What is the additional relevant information provided in this study when compared to the Choi et al or Kim et al study? (reference 8 and 12). this should be included in discussion.

Answer. 

I appreciate the critical comment. The data of Choi and Kim study were from both private hospital and university hospital, and the actual costs are different between private and university hospital. The purpose of this study was to suggest a practical reimbursement cost for FELD using a standardized clinical pathway while minimizing the use of uninsured surgical instruments. In this regards, we included patients in the non-profit tertiary referral hospital. Therefore, the homogeneity of data was the strength of this study. We clarified it in the limitation section as follows. 

Page 22, line 11

However, the main purpose of this study was to show the relevance of the costs incurred at hospitals for FELD. In this regard, we used homogeneous data from patients who underwent the same surgical procedure and were managed with the same clinical pathway in a single non-profit tertiary referral hospital. Although the number of patients was small, the surgical technique, medical costs, and postoperative management were homogeneous. In addition, no additional uninsured surgical materials such as a hemostatic agent or anti-adhesive agent were used, except for the electrode for FELD. We acknowledge that the small number of patients was the major weakness of this study, but the homogeneity may compensate for this limitation.

Line 29, page - correct FEED to FELD

Answer

I am sorry for typo error. It was corrected. 

General discussion on the method used for calculation of Cost per QALY should be added along with the comparison of other methods available.

Answer. 

I am sorry for not explaing the method of cost-utility analysis. Usually, economical aspect of surgical technique is evaluated by caculating the cost to increase a unit of health outcome measurement. The outcome measurement includes quality of life, pain, disability, reoperation or adverse event. The current study used a quality adjusted life year (QALY) for health outcome measurement, which was the most common measurement for cost-utility analysis. 

Page 19, line 3 

In evaluating a new surgical technique, the incurred cost to improve a unit in health outcome measurement is a common method in evaluating the technique under the health insurance system.[35,36] Utility measures varied between the EQ-5D and variations of the Short-Form Health Survey.[35] Effect measures varied widely between Visual Analogue Scale for pain, Neck Disability Index, Oswestry Disability Index, reoperation rates and adverse events.[35] The current study used EQ-5D for cost-utility analysis. The analysis showed the cost to improve a perfect 1-year quality of life per person (quality adjusted life year, QALY), which was the most common measurement for cost-utility analysis.

Limitations:

lines 17,18 page 17 are getting repeated and can be removed.

Answer. 

The repeated sentence was removed. 

lines 26,27 page 17, in the parent study of this study, there are patients who underwent open discectomy. Weren't operated in the same hospital? Comparing them for cost utility will give homogeneity.

Answer. 

I appreciate your comment. FELD was performed in our hospital and open discectomy was performed in the other hospital. Therefore, patients with open discectomy did not follow the clinical pathway of FELD. In addition, data of medical cost was not allowed to review without pre-planned study in the other hospital. As we noted, the main purpose of this study was to suggest a practical imbursement cost for FELD. We hope the current method would be appropriate for the purpose of this study.

---

## [Decision Letter · Decision Letter 2]

24 May 2023

PONE-D-21-21192R2Cost-utility analysis of endoscopic lumbar discectomy following a uniform clinical pathway in the Korean national health insurance systemPLOS ONE

Dear Dr. Chung,

Thank you for submitting your manuscript to PLOS ONE. After careful consideration, we feel that it has merit but does not fully meet PLOS ONE’s publication criteria as it currently stands. Therefore, we invite you to submit a revised version of the manuscript that addresses the points raised during the review process.

We look forward to receiving your revised manuscript.

Kind regards,

Sathish Muthu

Guest Editor

PLOS ONE

Journal Requirements:

Additional Editor Comments:

Dear Authors

I see that you have acknowledged the limitation pointed out in the discussion of the manuscript. You could also stress upon the need for a common surgical pathway for all the cases to make future assessments of evolving surgical techniques and methods feasible. Hence, I suggest acceptance of the present version following incorporation of the suggested minor edits to the limitation to stress the need for common surgical pathway.

Reviewers' comments:

Reviewer's Responses to Questions

**Comments to the Author**

1. If the authors have adequately addressed your comments raised in a previous round of review and you feel that this manuscript is now acceptable for publication, you may indicate that here to bypass the “Comments to the Author” section, enter your conflict of interest statement in the “Confidential to Editor” section, and submit your "Accept" recommendation.

Reviewer #2: All comments have been addressed

Reviewer #3: All comments have been addressed

2. Is the manuscript technically sound, and do the data support the conclusions?

Reviewer #2: Yes

Reviewer #3: Partly

3. Has the statistical analysis been performed appropriately and rigorously? 

Reviewer #2: Yes

Reviewer #3: Yes

4. Have the authors made all data underlying the findings in their manuscript fully available?

Reviewer #2: Yes

Reviewer #3: Yes

5. Is the manuscript presented in an intelligible fashion and written in standard English?

Reviewer #2: Yes

Reviewer #3: Yes

6. Review Comments to the Author

Reviewer #2: All comments were addressed so the paper can be accepted

All comments were addressed so the paper can be accepted

Reviewer #3: Dear Authors

I see that you have acknowledged the limitation pointed out in the discussion of the manuscript. You could also stress upon the need for a common surgical pathway for all the cases to make future assessments of evolving surgical techniques and methods feasible. Hence, I suggest acceptance of the present version following incorporation of the suggested minor edits to the limitation to stress the need for common surgical pathway.

7. PLOS authors have the option to publish the peer review history of their article (what does this mean?). If published, this will include your full peer review and any attached files.

Reviewer #2: **Yes: **Pravesh Gadjradj

Reviewer #3: **Yes: **Sathish Muthu

---

## [Author Response · Author response to Decision Letter 2]

25 May 2023

Author’s response to reviewers. 

Additional Editor Comments:

Dear Authors

I see that you have acknowledged the limitation pointed out in the discussion of the manuscript. You could also stress upon the need for a common surgical pathway for all the cases to make future assessments of evolving surgical techniques and methods feasible. Hence, I suggest acceptance of the present version following incorporation of the suggested minor edits to the limitation to stress the need for common surgical pathway.

Answer. 

Thank you for the positive comments for the revised manuscript. A standardization of surgical pathway will improve cost-utility of surgical treatment. This study showed the example and I appreciate the comment. We emphasized the importance of common surgical pathway like followings in the limitation section. 

Page 23, line 8

A common surgical pathway in care of patients and using surgical materials are required to incorporate FELD as one of appropriate surgical options.

Reviewer #2: All comments were addressed so the paper can be accepted

All comments were addressed so the paper can be accepted

Answer. 

Thank you for the positive comments for the revised manuscript. A standardization of surgical pathway will improve cost-utility of surgical treatment. This study showed the example and I appreciate the comment. We emphasized the importance of common surgical pathway like followings in the limitation section. 

Page 23, line 8

A common surgical pathway in care of patients and using surgical materials are required to incorporate FELD as one of appropriate surgical options.

Reviewer #3: Dear Authors

I see that you have acknowledged the limitation pointed out in the discussion of the manuscript. You could also stress upon the need for a common surgical pathway for all the cases to make future assessments of evolving surgical techniques and methods feasible. Hence, I suggest acceptance of the present version following incorporation of the suggested minor edits to the limitation to stress the need for common surgical pathway.

Answer. 

Thank you for the positive comments for the revised manuscript. A standardization of surgical pathway will improve cost-utility of surgical treatment. This study showed the example and I appreciate the comment. We emphasized the importance of common surgical pathway like followings in the limitation section. 

Page 23, line 8

A common surgical pathway in care of patients and using surgical materials are required to incorporate FELD as one of appropriate surgical options.

---

## [Decision Letter · Decision Letter 3]

31 May 2023

Cost-utility analysis of endoscopic lumbar discectomy following a uniform clinical pathway in the Korean national health insurance system

PONE-D-21-21192R3

Dear Dr. Chung,

We’re pleased to inform you that your manuscript has been judged scientifically suitable for publication and will be formally accepted for publication once it meets all outstanding technical requirements.

Kind regards,

Sathish Muthu

Guest Editor

PLOS ONE

Additional Editor Comments (optional):

Congratulations

Now i see that all the comments have be adequately addressed and now the manuscript is recommended for publication.

Reviewers' comments:

Reviewer's Responses to Questions

**Comments to the Author**

1. If the authors have adequately addressed your comments raised in a previous round of review and you feel that this manuscript is now acceptable for publication, you may indicate that here to bypass the “Comments to the Author” section, enter your conflict of interest statement in the “Confidential to Editor” section, and submit your "Accept" recommendation.

Reviewer #3: All comments have been addressed

2. Is the manuscript technically sound, and do the data support the conclusions?

Reviewer #3: Yes

3. Has the statistical analysis been performed appropriately and rigorously? 

Reviewer #3: Yes

4. Have the authors made all data underlying the findings in their manuscript fully available?

Reviewer #3: Yes

5. Is the manuscript presented in an intelligible fashion and written in standard English?

Reviewer #3: Yes

6. Review Comments to the Author

Reviewer #3: Dear Authors

Congratulations on the manuscript now being accepted for publication in our esteemed journal.

7. PLOS authors have the option to publish the peer review history of their article (what does this mean?). If published, this will include your full peer review and any attached files.

Reviewer #3: **Yes: **Sathish Muthu

---

## [Editor Report · Acceptance letter]

7 Jun 2023

PONE-D-21-21192R3 

Cost-utility analysis of endoscopic lumbar discectomy following a uniform clinical pathway in the Korean national health insurance system 

Dear Dr. Chung:

I'm pleased to inform you that your manuscript has been deemed suitable for publication in PLOS ONE. Congratulations! Your manuscript is now with our production department. 

Kind regards, 

on behalf of

Dr. Sathish Muthu 

Guest Editor

PLOS ONE